# The Influence of Microstructure on Abrasive Wear Micro-Mechanisms of the Claddings Produced by Welding Used in Agricultural Soil

**DOI:** 10.3390/ma13081920

**Published:** 2020-04-19

**Authors:** Aleksandra Królicka, Łukasz Szczepański, Łukasz Konat, Tomasz Stawicki, Piotr Kostencki

**Affiliations:** 1Faculty of Mechanical Engineering, Wroclaw University of Science and Technology, Wybrzeże Wyspiańskiego 27, 50-370 Wroclaw, Poland; lukasz.szczepanski@pwr.edu.pl (Ł.S.); lukasz.konat@pwr.edu.pl (Ł.K.); 2Department of Engineering of Renewable Energy Sources, Faculty of Environmental Management and Agriculture, West Pomeranian University of Technology in Szczecin, Papieza Pawla VI Street 1, 71-459 Szczecin, Poland; tomasz.stawicki@zut.edu.pl (T.S.); Piotr.Kostencki@zut.edu.pl (P.K.)

**Keywords:** wear, Fe-Cr-C-Nb alloys, hardfacing, M7C3, NbC, white cast iron, agricultural soil

## Abstract

Claddings produced by welding are commonly used to increase the durability of the working elements of agricultural tools. The working conditions that occur during the cultivation of agricultural soil determine the wear intensity (different soil fractions, biological, and chemical environment). It was found that the tested claddings (Fe-Cr-C-Nb system) is characterized by three different layers: hypereutectic (layer I), near eutectic (layer II), and hypoeutectic (layer III). In layer I, micro-cracking and spalling of hard and brittle primary M_7_C_3_ carbides resulted in micro-delamination under the impact of larger soil fractions, which increased the wear intensity. Due to the lower fraction of primary M_7_C_3_ carbides in layer II, the share of micro-delamination was less significant in comparison to layer I. It was found that niobium carbides are firmly embedded in the matrix and effectively inhibit wear intensity in layer I and layer II. Layer III contained austenite dendrites, a refined eutectic mixture, and also NbC. In this layer, cracks (caused the unfavorable eutectic mixture morphology) were found in the interdendritic spaces at the worn surface. After the penetration of the cladding, there was a "wash-out effect", which resulted in a significant reduction in the durability of the working elements due to abrasive wear.

## 1. Introduction

Structural elements of machines operating in soil are particularly exposed to heavy abrasive wear and moderate impact loads (in analyzed working conditions). The mechanisms of wear of elements working in soil show a complex nature, including primarily abrasive wear, which is a result of the many variables determining the properties of the abrasive fraction, such as: the size of soil particles, soil granulometric composition [1,2], soil humidity [3,4,5,6], or cultivation speed [7]. Due to the reduction of costs associated with the prolonged use of the working parts of agricultural tools, material selection, which is perfectly adapted to the requirements of the tribological system, is very important. Research has been published that describes the mechanisms of abrasive wear of commonly used materials, such as martensitic steels with boron [8,9], cemented-carbides [10,11], and claddings produced by welding [12,13,14,15].

This article focuses on a detailed analysis of the heterogeneous structure of claddings produced by welding and its impact on soil wear micro-mechanisms. Buchely et al., in work [13], reports that the microstructure and number of weld seams plays a key role in the intensity of abrasive wear. Hard M_7_C_3_ carbides have a particularly significant effect on increasing wear resistance [16,17]. However, the presence of large and brittle primarily solidified chromium carbides from the melt may be the cause of intensified wear, which can be seen in the results of their break out from the matrix in highly loaded working conditions [18,19]. Moreover, Chung et al. [20], in laboratory wear behavior tests (pin-on-disc), found that the presence of dispersive niobium carbides increases abrasive wear resistance when compared to Fe-Cr-C alloys. In addition, the presence of fine and hard niobium carbides also causes the refinement and shape of primary M_7_C_3_ carbides [21], which improves their overall wear resistance under dynamic loads. In work [22], it was found that a content of about 17.0 wt % Cr, about 3.0 wt % C and about 3.0 wt % Nb results in the best combination of impact strength and abrasive wear resistance (an increase of about 30% when compared to Fe-Cr-C white cast iron). A similar welding alloy was also analyzed in this article. On the other hand, Fiset et al. [23] found that optimal mechanical properties were achieved in Fe-C-Cr-Nb white cast iron with a 2 wt % Nb content. Of course, it should be highlighted that these were attempts to increase the abrasive wear resistance of Fe-Cr-C-Nb alloys under certain working conditions.

In previous work [24], the influence of a construction solution (materials and geometry) on the intensity of cultivator coulter wear during soil cultivation was assessed. The heterogeneous microstructure of the claddings produced by welding and the complex wear mechanisms were identified, which were the reason for developing the analysis of their microstructure in qualitative and quantitative terms. A detailed analysis of the microstructure of claddings produced by welding from the Fe-Cr-C-Nb system, together with the assessment of their wear micro-mechanisms in soil, can provide prospects for the future design of the hardfacing process in terms of maximizing their wear resistance under specific soil conditions. The assessment of the dominant wear micro-mechanisms of individual layers of claddings supplements the current considerations on abrasive wear resistance, especially with regards to tests performed in real and difficult working conditions that prevail during soil cultivation.

## 2. Materials and Methods 

The claddings produced by welding used in cultivator coulters that are also reinforced with cemented-carbide plates were tested (Figure 1). In previous work [24], various construction solutions of coulters were analyzed in the context of their wear resistance in specific soil conditions. The soil physicochemical properties and soil granulometric composition, working conditions, and method of attachment in the cultivator were presented in previous work.

Observations of the microstructure were carried out using light and electron microscopy in the etched state. The Nikon Eclipse MA200 light microscope (Tokyo, Japan) with a Nikon DS-Fi5 CCD camera, and the JEOL JSM-6610A scanning microscope (Tokyo, Japan) were used. SEM observations were performed using 20 kV accelerating voltage and 10 mm working distance. Material contrast (BSE detector—back-scattered electrons) was used in the studies of microstructure using SEM.

Samples of claddings produced by welding for the microstructure tests were cut in their cross-section. The preparation of the samples included grinding on abrasive papers and polishing with diamond pastes, ending with diamonds not exceeding 1 μm. The samples were then electrolytically etched using H_2_CrO_4_ electrolyte.

Observations of wear mechanisms were made using scanning microscopy (JEOL, JSM-6610A, Tokyo, Japan) and an SE and BSE detector. The observations were carried out on both the surface of the worn claddings and on the cross-section. 

The comparative analysis of the chemical composition of the individual layers of claddings was carried out by the EDX method using an energy-dispersed x-ray spectrometer JEOL JED-2300 (Tokyo, Japan) coupled with the JEOL JSM-6610A scanning electron microscope (Tokyo, Japan). Segregation of chemical elements (excluding carbon) was performed by surface microanalysis using the same test area. The distribution of elements in the layers of the claddings was made by applying mappings for the elements: Cr, Fe, and Nb. Quantitative analysis of the structure components was performed by graphical image editing with the ImageJ program (Version 1.51w, National Institutes of Health, Federal Government, US), using various phase contrasts obtained by using a BSE detector (material contrast).

Carbide identification was performed using Transmission Electron Microscopy and the SAED (Selected Area Diffraction Pattern) method. The tests were carried out on a Hitachi H-800 transmission microscope (Tokyo, Japan). Samples of the claddings were cut into slices with a thickness of 500 μm using the electroerosion method. The samples were then mechanically thinned to a thickness of 60 μm. Discs with a diameter of ø3 mm were mechanically cut from the thinned samples. Then, electrochemical polishing was performed on the Struers TenuPol (Cleveland, OH, USA) device and ion polishing was carried out using the GATAN DuoMill device (Pleasanton, CA, USA).

The hardness of the claddings was determined by the Vickers method on a MMT-X3 hardness tester (Matsuzawa, Akita, Japan). A load of 9.807 N was applied for 15 s. In order to determine the changes in hardness of individual layers, the tests were performed in a line from the surface to the base of the material. The first measurement was made at a distance of 0.1 mm from the cladding surface, with subsequent measurements being made at a distance of 0.25 mm.

## 3. Results and Discussion

### 3.1. Materials Characterization

#### 3.1.1. Macroscopic Wear Symptoms and Hardness Distribution

The claddings produced by welding of the cultivator coulter, with a chemical composition of 4.32C-0.37Mn-1.04Si-16.79Cr-3.92Nb-Fe(bal.), was tested (Figure 2). It is one of the Fe-Cr-C-Nb type commercial welding alloys used to increase abrasive wear resistance. In previous studies conducted during soil cultivation, it was found that hardfacing effectively contributed to reducing the wear intensity of the working parts of agricultural tools [24]. During tests under real operating conditions, atypical symptoms of the wear of pad-welded elements were noticed. It was found that the claddings showed heterogeneous wear processes, which was evidenced by numerous examples of visible macro-delamination (Figure 3). A particular intensification of wear occurred at the border of the cladding and the base material (“wash-out” effect). On the basis of cultivator coulters after soil exploitation, an attempt was made to explain the mechanisms of wear in relation to their microstructure.

Figure 4 shows the hardness distribution, which confirmed that microstructural changes in claddings affect their hardness. Based on changes in hardness, as well as their varied microstructure, three layers were distinguished: layer I, layer II, and layer III. In the following sections of this article, references to the layers are in accordance with Figure 4. The hardness for individual layers remains approximately constant and changes significantly after exceeding the layer boundary. The highest hardness (approximately 780 HV1) was obtained for layer II, where primary M_7_C_3_ carbides, dispersive niobium carbides and eutectic (γ + M_7_C_3_) occurred. This increase may also be associated with a higher dispersion of eutectic when compared to layer I. However, the lowest hardness (approximately 670 HV1) was obtained for layer III, which can be explained by the occurrence of dendritic austenite with a relatively low hardness but high ductility. The lack of primary M_7_C_3_ carbides in this layer could also have affected the hardness reduction. Detailed microstructure analysis of all the layers is described in the following sections.

#### 3.1.2. Identification of Microstructure Components 

In order to unequivocally identify the structure of components present in the tested cladding weld, examinations were carried out using Transmission Electron Microscopy and the SAED (Selected Area Electron Diffraction Pattern) method. Figure 5, Figure 6, Figure 7 and Figure 8 show all the occurring structure components: primary and eutectic (Fe,Cr)_7_C_3_ carbide, primary NbC carbide, and austenite. 

It is well known that primary M_7_C_3_ carbide with a needle or fragmented morphology occurs in high-chromium cast irons (among others, in works [13,16,17,18,19,20,21,22,23]). Chromium carbide with a fragmentary, hexagon morphology was selected for identification (Figure 5a). In the case of M_7_C_3_ carbide, the presence of stacking faults and micro-twins was found. In addition, it was also found that the angle between the bands is rotated with respect to each other by 120° and 240° degrees, which indicates that the observed carbide has a polycrystalline structure [25,26].

The solubility of niobium in austenite and M_7_C_3_ carbide is very low, so most of the niobium present in the alloy is in the form of MC carbide [27]. In all the layers, the niobium carbide morphology was polygonal or nodular. An example of the niobium carbide is shown in Figure 6. Based on the diffraction pattern solution (Figure 6b), the MC type of carbide was identified.

An austenitic matrix was identified between the primary carbides M_7_C_3_ (Figure 7). The austenite in high-chromium cast iron may partially transform into martensite [28]. The Kurdjumov–Sachs Orientation Relationship between ferrite and austenite, which indicates the presence of martensite, was not found (Figure 7c).

In the area of the eutectic mixture, the presence of eutectic M_7_C_3_ carbide was found (Figure 7), which also occurs in high chromium cast iron [29,30]. The presence of secondary M_23_C_6_ carbide in the eutectic area was not confirmed. This type of carbide in the dendritic area occurs during prolonged heating of high-chromium cast iron at a temperature of 900–1100 °C [28]. Due to its hardness, M_7_C_3_ carbide increases the abrasive wear resistance, but at the same time reduces toughness. However, morphology and orientation also significantly affect these properties [29]. The eutectic carbides in layer III of the tested cladding were characterized by a needle-like morphology with a width generally less than 100 nm (Figure 8a). In addition, eutectic carbides mostly showed sharp edges, which indicates a stress concentration and a significant notch effect. Thus, the fragmentation, refinement, and morphology of the M_7_C_3_ carbides in eutectics may be evidence of good abrasive wear resistance, but reduced toughness.

#### 3.1.3. Chemical Elements Distribution

Figure 9 shows the sum of EDX mappings for Fe, Cr, and Nb for the three layers. In layers I and II, there were both the primary and the eutectic chromium-iron carbide M_7_C_3_ type. The mapping confirmed that both Cr and Fe are dissolved in M_7_C_3_. The morphology of eutectics (γ + M_7_C_3_) is different for all the layers. In layer I (Figure 9a), an irregular, coarse eutectic without clear lamellar morphology is visible. In layer II (Figure 9b), however, the eutectic exhibits a clearly lamellar and finer morphology, and in layer 3 (Figure 9c) the eutectic is significantly refined and distributed in the interdendritic spaces of the austenite. In addition, it was previously found (Figure 9a) that the eutectic in this layer is characterized by strong refinement (width of eutectic carbide less than 100 nm) and unfavorable morphology. All the layers also contained the primary niobium carbide with a polygonal or nodular morphology. Polygonal NbC primarily occurred in layer I, while in layer II, nodular NbC predominated. In contrast, layer III mainly contained nodular NbC. Niobium carbide with “chinese-script” morphology, reported in works [31,32], was not identified. 

No segregation of chemical elements (excluding carbon) was found in the first two layers of the cladding (Figure 10 and Table 1). A slight decrease in Cr, Nb, and Si was found in the case of the last layer located directly at the base material, which can be explained by their diffusion into the base material during the hardfacing process.

#### 3.1.4. Microstructure Components Distribution

As in others works [20,33], it was found that primary needle chromium carbides are oriented according to the heat flow direction. The amount of primary needle carbides M_7_C_3_ is definitely the highest in layer I (Figure 11a and Table 2). It was found that in addition to needle chromium carbide, there are also fragmented chromium carbides with a shape similar to a hexagon, whose proportion and fragmentation increases as a function of the distance from the cladding surface to the boundary of layer II (Figure 11b). According to [20,21,22,23,34], refinement of primary carbides M_7_C_3_ is caused by the presence of niobium carbides, which constitute a barrier to their free growth and react with carbon (over iron or chromium). Once NbC has formed, there will be less carbon remaining in the melt to react. For individual layers, a change in dispersion and distribution of micrstructure components was also visible. Niobium carbides form clusters in the first layer, the intensity of which decreases towards the last layer. In layer III, niobium carbides are distributed in interdendritic spaces and are generally evenly distributed. The amount of fragmented, polygonal M_7_C_3_ primary carbides is the largest in the second layer. Primary M_7_C_3_ carbides in layer III were not identified (Figure 11c and Table 2). For this reason, it was found that the claddings produced by welding used in the experiment consisted of layers: layer I—hypereutectic structure, layer II—generally a eutectic structure, and layer III—a hypoeutectic structure.

### 3.2. Mechanisms of Abrasive Wear

#### 3.2.1. Topography of the Worn Surface

Various wear mechanisms under real working condition of the specified layers of the claddings produced by welding were identified (Figure 12). Based on the microstructural analysis, it was found that the welds were hypereutectic (layer I), near eutectic (layer II), and hypoutectic (layer III). According to [35], hypereutectic alloys exhibit better wear resistance than hypoeutectic alloys due to the presence of coarse primary chromium carbides and their higher hardness. The highest abrasive wear resistance of layer 1 is fulfilled for the dusty soil fraction. However, during operation in soil, stones also affect the coulters, which introduces impact-working conditions. In the case of strong, dynamic loading by stones, coarse primary M_7_C_3_ carbides break out of the eutectic matrix, which causes micro-delamination (Figure 12a). Pits caused by cracked and spalled chromium carbides intensify further wear and thus facilitate the penetration of abrasive medium in the eutectic matrix. It was found that pits have micro-cracks in the area of the eutectic mixture (Figure 13), which contribute to increased wear in this area. In layer I, in addition to micro-delamination, scratches associated with the microcutting mechanism also occurred (Figure 12a). Cracks related to hardfacing technology were also found. In layer II, pits caused by primary M_7_C_3_ carbide spalling were also identified. However, due to their higher dispersion and lower fraction, when compared to layer I, they did not cause increased micro-delamination. This layer (Figure 12b) was dominated by the micro-cutting mechanism (scratches), and the niobium carbides firmly embedded in the eutectic matrix were visible, which, similar to work [36], inhibited abrasive wear. The favorable effect of NbC on abrasive wear was also confirmed in [20,21,22,23,37]. In both layers I and II, it was found that primary M_7_C_3_ carbides are prone to spalling, and that NbC are firmly embedded in the matrix, as seen in Figure 14. Niobium carbides are removed from the cladding material only after removal of the matrix around them, which weakens their embedding. However, in the case of layer III, it was found that the dominating mechanisms are micro-cutting (scratches) and ploughing (grooves) (Figure 12c). Areas showing plastic deformation were also identified, which is characteristic for hypoeutectic alloys [35]. At the worn surface in this layer, the presence of pit caused by the chipping of niobium carbides was also found. Niobium carbides in this layer were not strongly embedded in the matrix and did not inhibit abrasive wear as in the case of layers I and II. At the border of the base material and layer III, there was a “wash-out” effect characterized by intensified wear in this area (Figure 12d). This can be explained by the significant difference in hardness and mechanical behaviors of both materials (claddings and steel). In addition, austenite was present at the border of the weld and base material, which also resulted in higher wear in this area. It can therefore be concluded that after complete penetration of the cladding weld, the coulters were subjected to significantly increased abrasive wear and the durability of the element was reduced.

#### 3.2.2. Cross-section of the Worn Surface 

In works [38,39], it was found that the worn surface of claddings produced by welding can be hardened by inducing martensitic transformation from austenitic areas, which allows their resistance to abrasive wear to be increased. In order to assess the hardened layer, metallographic samples after wear were prepared on the cross-section (Figure 15). Observation of layer 1 did not reveal a macroscopic hardened layer at the worn surface. However, cracks of primary M_7_C_3_ carbides (Figure 15a) are visible, which are probably caused by the impact of a larger fraction of soil. In layer II, the presence of a thin hardened layer was found (Figure 15b). The highest share of the hardened layer was found in layer III (Figure 15c), which was also confirmed for hypoeutectic alloys [38]. However, at the worn surface, numerous cracks are visible in the interdendritic areas. Therefore, it can be hypothesized that when working in soil, the hardening created by inducing the martensitic transformation causes cracks resulting from the simultaneous interaction of transformations, residual stress, and dynamic loads under these working conditions. In this case, hardening the worn surface will adversely affect abrasive wear, and the occurring cracks will cause wear intensification. In addition, the unfavorable morphology of strongly refined eutectics (Figure 8a) in this layer caused stress concentrations and cracks in the interdendritic area under the influence of induced martensitic transformation and fatigue loads. For this reason, the wear rate was higher than in the case of the previous two layers. Combining the results obtained from the observation of the topography (Figure 12c) and cross-section (Figure 15c), it can be concluded that the effect of wear inhibition by niobium carbide is not noticeable due to the presence of microcracks at the worn surface. The general topography of the worn surface of layer III, in which many pits, scratches, and plastic deformation are visible, is due to the presence of cracks and a high proportion of the plastic austenitic matrix, which has partially undergone martensitic transformation.

## 4. Conclusions

Based on the analysis of wear mechanisms in relation to the microstructure of claddings produced by welding used for soil cultivation, the following conclusions were formulated:Macroscopic wear mechanisms of the cultivator coulters confirmed the complex wear micro-mechanisms of claddings produced by welding. On the border of the base material (steel) and the initial layer (III) of the claddings, intensified wear occurred (“wash-out” effect).The tested claddings consisted of three microstructurally different layers. Based on the analysis of the fraction of structure components, hardness measurements and microanalysis of the chemical composition, it was found that these layers can be classified as hypereutectic, approximately eutectic, and hypoeutectic alloys.The claddings layers were characterized by various wear mechanisms. In layer I (hypereutectic), the micro-cutting mechanism and delamination due to chipping of primary (Cr,Fe)_7_C_3_ carbide dominated. Layer II (approximately eutectic) was dominated by the micro-cutting mechanism. However, in layer 3 (hypoeutectic), scratches and ploughing with the presence of plastically deformed areas and numerous pits were observed.It was found that in layer I there was no macroscopic hardening of the worn surface, while in layer II a narrow hardened layer was identified. The largest layer of hardening was observed in layer III. However, the effect of the residual stress of martensitic transformation, the impact of high loads of a larger soil fraction, the fatigue nature of loads, and the unfavorable morphology (sharp edges of needles) of the strongly eutectic refinement caused the occurrence of microcracks in interdendritic spaces. The cracks located at the worn surface contributed to the intensified wear of this layer.The results, obtained in real working conditions (soil cultivation), partly confirm previous research, which explains the need to validate laboratory results with results obtained during work in specific conditions.Generally, in the case of the analyzed working conditions and microstructures of the tested claddings produced by welding, it can be concluded that the most advantageous mechanical behavior in the context of wear resistance in tested working conditions was demonstrated by layer II (approximately eutectic).

## Figures and Tables

**Figure 1 materials-13-01920-f001:**
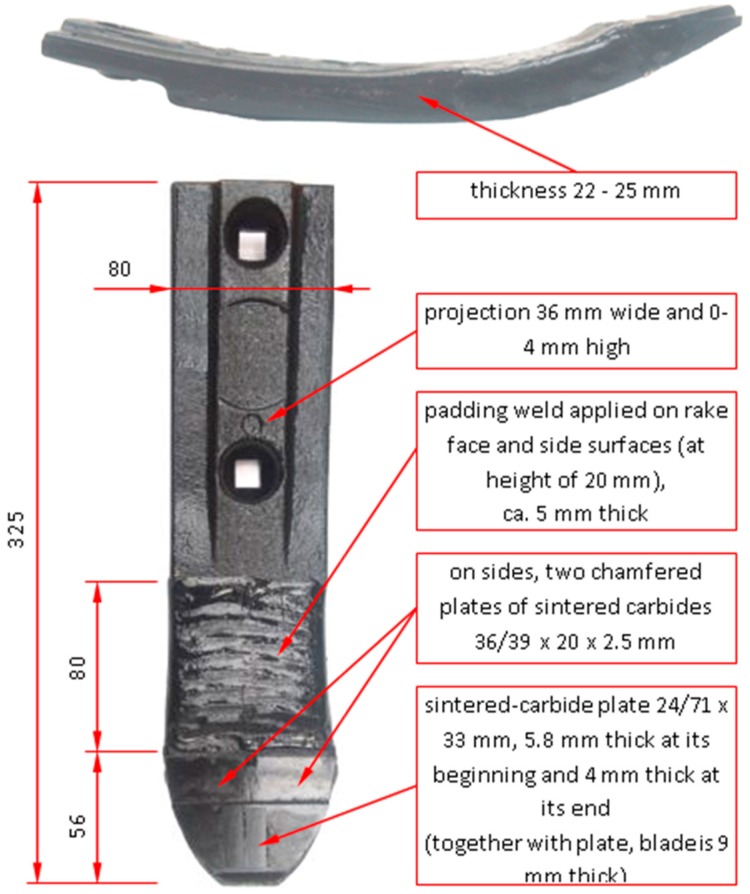
Geometry and structure of the coulters used in this research [24].

**Figure 2 materials-13-01920-f002:**
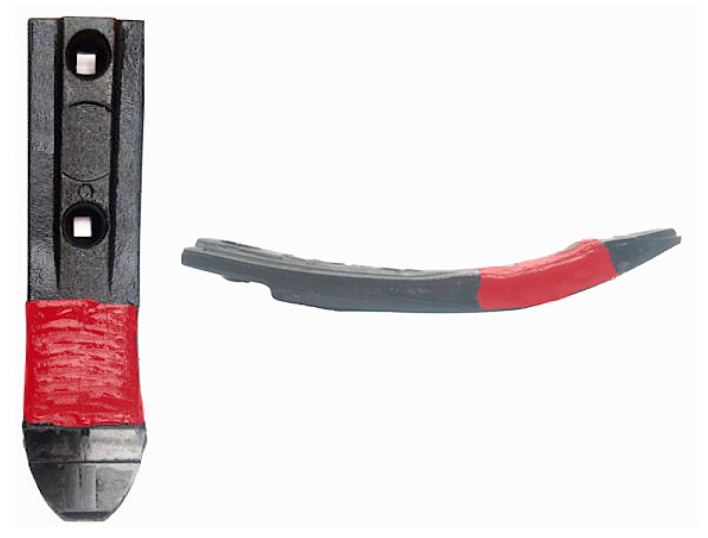
Location of the tested claddings produced by welding on the cultivator coulter.

**Figure 3 materials-13-01920-f003:**
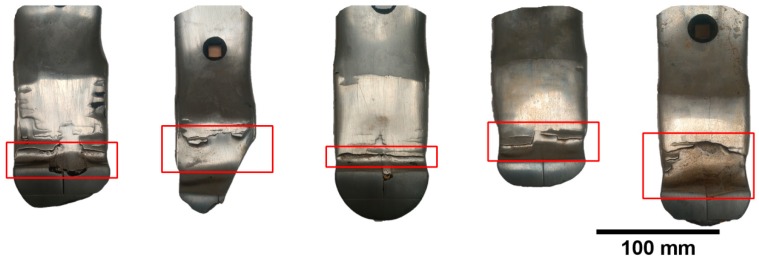
View of the claddings area of cultivator coulters after working in soil. Macroscopic delamination on the surface of the claddings (“wash-out” effect), visible inside frames.

**Figure 4 materials-13-01920-f004:**
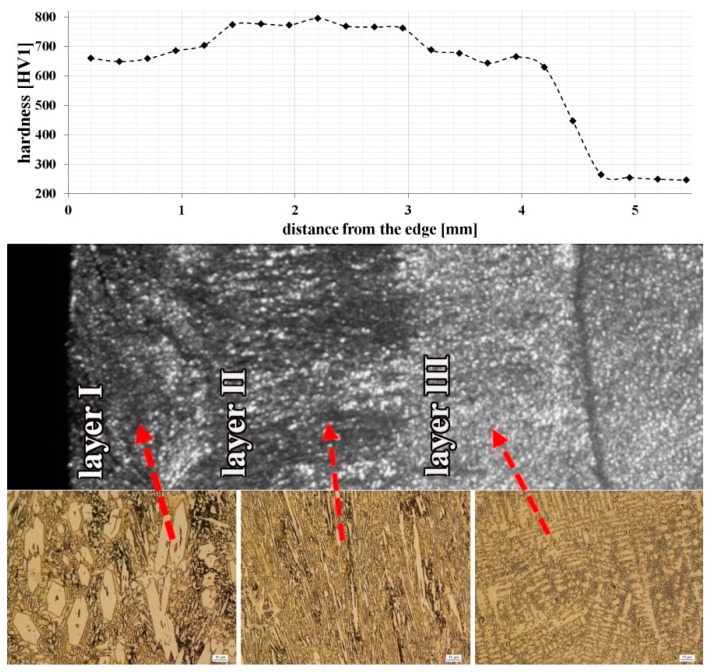
Hardness distribution of the tested claddings produced by welding with distinguished zones characterized by a different microstructure.

**Figure 5 materials-13-01920-f005:**
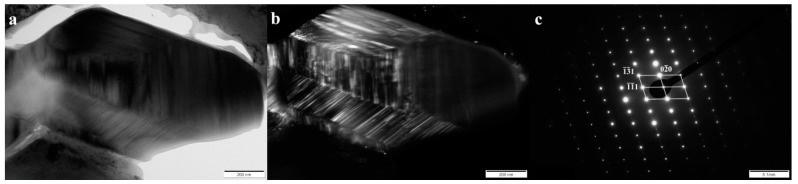
(**a**) Bright field image of the primary M_7_C_3_ carbide in layer I. (**b**) Dark field image obtained from reflection (0–20) shown in (**c**). (**c**) Selected area diffraction pattern (SAED) of the primary M_7_C_3_ carbide [101] shown in (**a**), along with the solution. Transmission electron microscope, 150 kV.

**Figure 6 materials-13-01920-f006:**
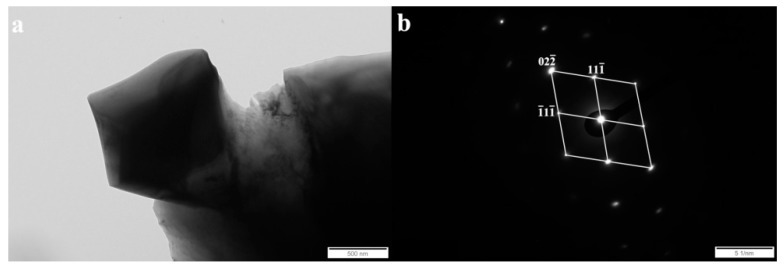
(**a**) Bright field image of the NbC carbide in layer I. (**b**) Selected area diffraction pattern (SAED) of the primary NbC [011] carbide shown in Figure 6a, along with the solution. Transmission electron microscope, 150 kV.

**Figure 7 materials-13-01920-f007:**
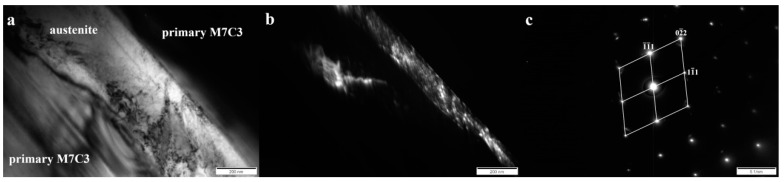
(**a**) Bright field image of austenite between primary M_7_C_3_ carbides in layer II. (**b**) Dark field image obtained from reflection (1 $\bar{1}$
$\bar{1}$) from austenite—shown in (**c**). (**c**) Selected area diffraction pattern (SAED) of austenite (011) shown in (**a**), along with the solution. Transmission electron microscope, 150 kV.

**Figure 8 materials-13-01920-f008:**
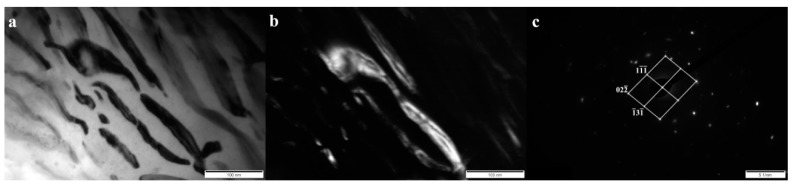
(**a**) Bright field image of eutectic (γ + M_7_C_3_) in layer III. (**b**) Dark field image obtained from reflection (1-1-1) from the eutectic M_7_C_3_ carbide—shown in (**c**). (**c**) Selected area diffraction pattern (SAED) of the area shown in (**a**), along with the solution. Eutectic carbide M_7_C_3_ [211] was found. Transmission electron microscope, 150 kV.

**Figure 9 materials-13-01920-f009:**
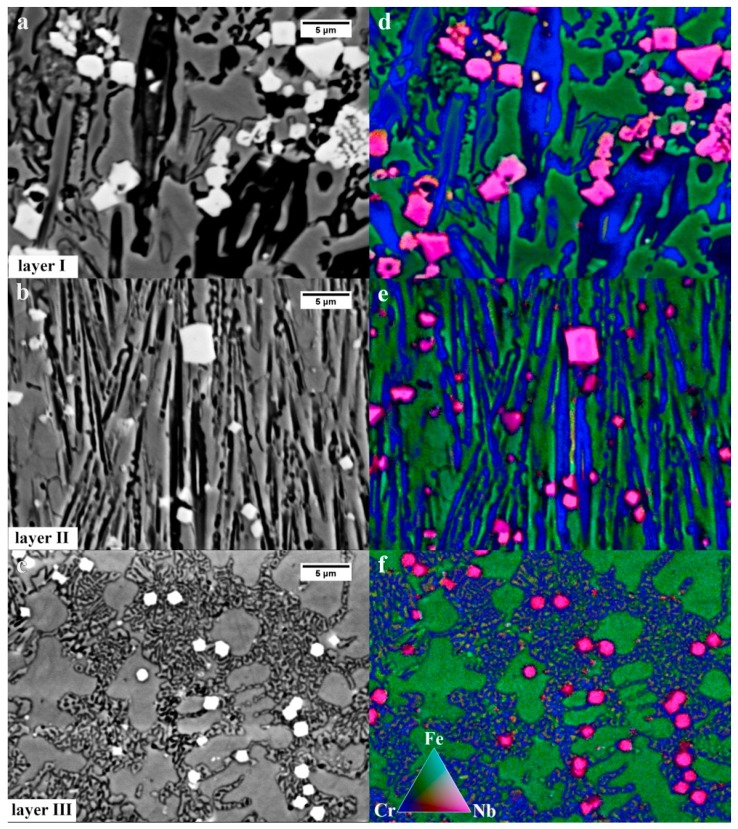
(**a**) Chemical elements distribution. (**a**–**c**) Microstructures obtained with the material contrast- BSE detector. (**d**–**f**) Sum of EDX mappings for Fe, Cr, and Nb. Scanning electron microscopy.

**Figure 10 materials-13-01920-f010:**
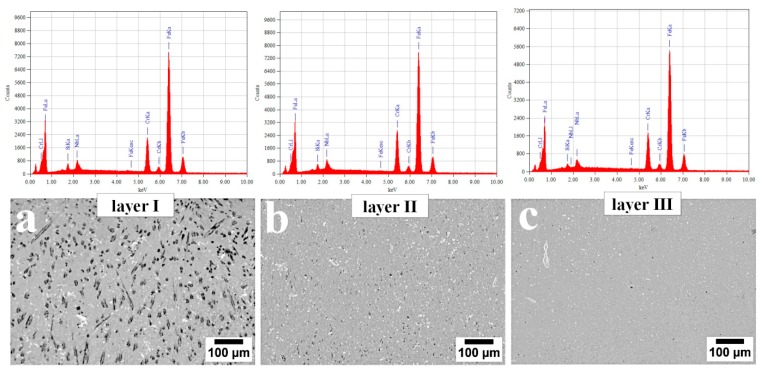
Energy-dispersive X-ray spectroscopy (EDX) obtained from distinguished areas of the cladding produced by welding. (**a**) Layer I; (**b**) Layer II; (**c**) Layer III. Scanning electron microscopy.

**Figure 11 materials-13-01920-f011:**
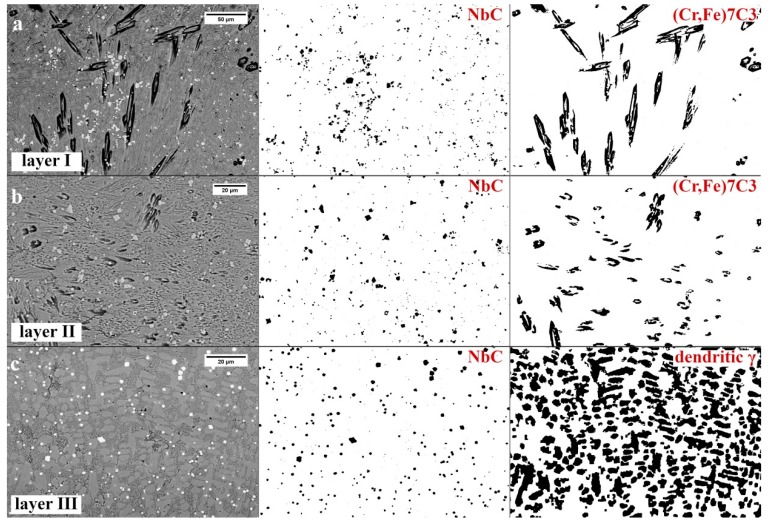
Graphic quantitative fraction of structure components. (**a**) Reference image for layer I, niobium carbides, primary chromium carbides, (**b**) Reference image for layer II, niobium carbides, primary chromium carbides, (**c**) Reference image for layer III, niobium carbides, dendritic austenite. Scanning electron microscopy.

**Figure 12 materials-13-01920-f012:**
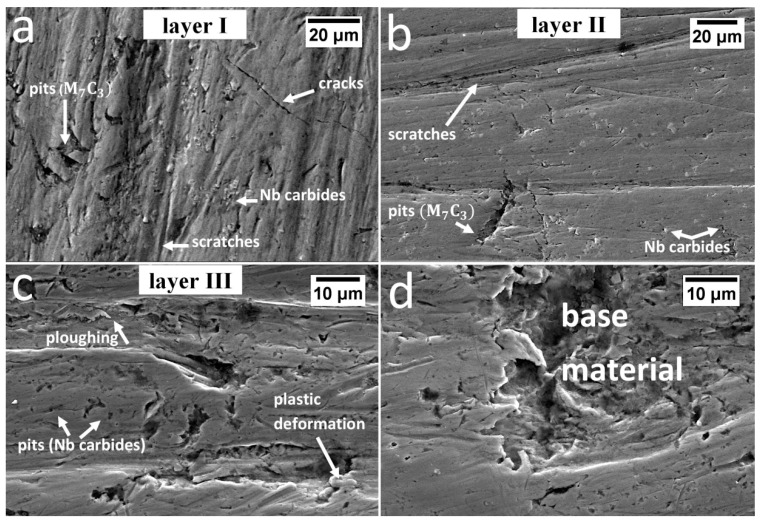
Wear mechanisms of the tested pad-weld. (**a**) Layer I: visible micro-delamination caused by primary M_7_C_3_ carbide spalling, scratches (micro-cutting mechanism), Nb carbides, and technological cracks. (**b**) Layer II: visible pits caused by primary M_7_C_3_ carbide spalling, firmly embedded niobium carbide, and scratches (micro-cutting mechanism). (**c**) Layer III: visible areas of plastic deformation, ploughing, and pits caused by niobium carbide chipping. (**d**) Layer III/Base material: Visible boundary between the pad-weld and base material—“wash out” effect. Intensified material wear. Scaninng electron microscopy.

**Figure 13 materials-13-01920-f013:**
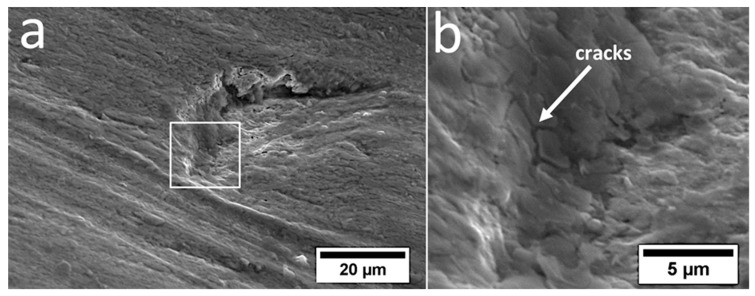
(**a**) Pit caused by primary chromium carbide spalling. (**b**) Visible microcracks in the pit area. Scanning electron microscopy.

**Figure 14 materials-13-01920-f014:**
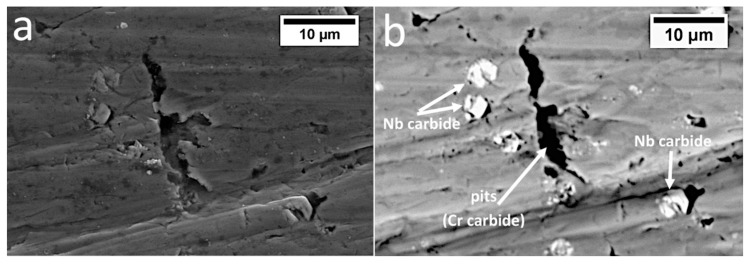
Area of layer I. (**a**) Topographic contrast (SE detector). (**b**) Material contrast (BSE detector). Visible pits caused by chromium carbide spalling and firmly embedded niobium carbides. Scanning electron microscopy.

**Figure 15 materials-13-01920-f015:**
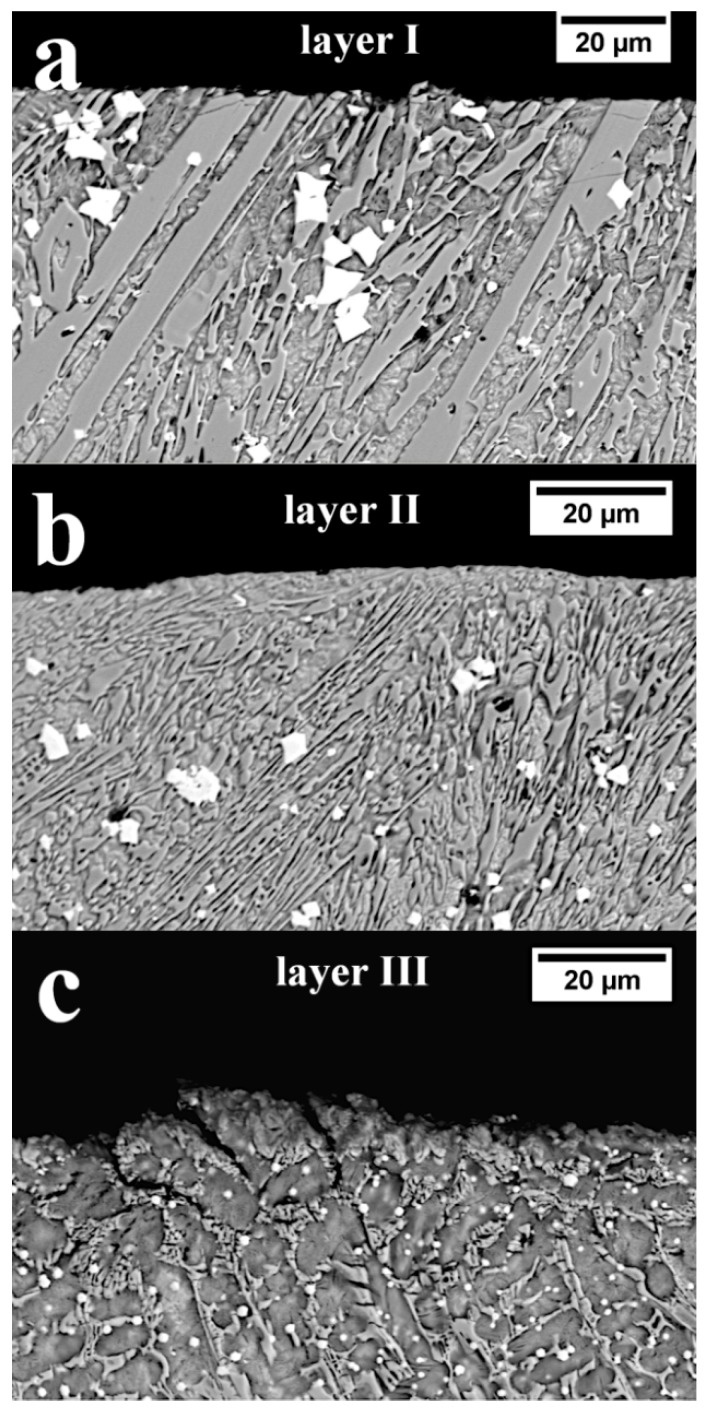
Cross-section of claddings produced by welding after operating in soil. (**a**) Layer I: no visible hardened layer during wear. Visible cracks of primary M_7_C_3_, indicated with an arrow (**b**) Layer II: visible slight hardened layer during wear. (**c**) Layer III: visible hardened layer and cracks in interdendritic spaces. Scanning electron microscopy.

**Table 1 materials-13-01920-t001:** Quantitative results of the chemical composition microanalysis from the areas shown in Figure 10.

	Chemical Composition (wt. %)			
Layer	Cr	Nb	Si	Fe
I	15.85	4.38	1.35	Bal.
II	16.04	4.48	1.16	Bal.
III	14.86	3.40	0.71	Bal.

**Table 2 materials-13-01920-t002:** Quantitative results of the fraction of structure components obtained on the basis of the graphic edition of the images shown in Figure 11.

	Volume Fraction [%]			
Layer	(γ + M_7_C_3_)	Primary M_7_C_3_	Primary NbC	γ (Dendritic)
I	86.5	10.4	3.1	-
II	92.3	5.4	2.3	-
III	57.5	-	2.8	39.7

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
