# Peer review of "The Influence of Microstructure on Abrasive Wear Micro-Mechanisms of the Claddings Produced by Welding Used in Agricultural Soil"

_materials, 2020, doi:10.3390/ma13081920_

Round 1
Reviewer 1 Report
This paper is a thorough investigation on the abrasive wear micro-mechanisms for cultivator coulters with welded hard-facing cladding. The paper increases the understanding of the wear mechanisms and how it is related to the variation of the cladded microstructure. Therefore provides important knowledge to the scientific community on how the variation of the microstructure of Fe-Cr-C-Nb claddings directly affects the wear rate. Such understanding can assist in improving the cladding process to maximise the preferred microstructure and increase part life, so this work also has a clear industrial significance and impact.
The paper was well written and results/discussions were carefully laid out in an easy to follow structure. The Introduction highlighted the reason for their work and provided sufficient background detail and references to set up the research. The authors showed a sound understanding of the microstructure and carbide formation in the identification section. The images were also of high quality. This allowed the authors’ to provide a clear and good explanations of how the microstructure contributed to the wear behaviour. The effort made to separate the phases (Fig 11), made it much easier for the reader to interpret the microstructures. Through the interpretation of the data in the paper and use of referenced research, strong conclusions were made.
There are a few minor points for correction;
P1 L22 refinement should be refined
P2 L85 layers of claddingsds was … should be claddings were…
P6 L184 Fig 7a is referred to when describing layer III, which is Fig 8a.
P7 l197 mappings should be mapping
P7 L198-200 Fig 8, in brackets, should be Fig 9a, b & c
P7 L203 space in wrong place for 100 nm
P8 L233 components ‘is’ should be ‘was’
P9 L258 operating to operation
P10 L288 Fig 12. Add space before (a). tehcnological should be technological
P11 L296 SE detector
P12 L 327 Fig 14. Should be Fig 15. Also, Fig 15a would benefit from arrows highlighting the cracks.
P12 L336. The “last layer” of the cladding is a little confusing, I suggest changing it to ‘initial layer’.
P13 L346 change is to was.
P13 L347 change to: The largest layer of hardening was…
P13 L356-360 Final conclusion is fine, so can delete the last sentence.
One point for the Authors to consider is the effect of the HAZ of the base steel may also be contributing to the wash out effect at the interface. I.e. Coarse Grain HAZ may have reduced toughness. As the research is based on interpretation of the cladding material, I don’t see it necessary to investigate this and it is also hard to eliminate the HAZ.
Author Response
Dear Reviewer,
Thank you for your review of our manuscript and a few important comments. We hope that with their help we were able to improve the quality of the article.
1. P1 L22 refinement should be refined; P2 L85 layers of claddingsds was … should be claddings were…; P6 L184 Fig 7a is referred to when describing layer III, which is Fig 8a.; P7 l197 mappings should be mapping; P7 L198-200 Fig 8, in brackets, should be Fig 9a, b & c; P7 L203 space in wrong place for 100 nm; P8 L233 components ‘is’ should be ‘was’; P9 L258 operating to operation; P10 L288 Fig 12. Add space before (a). tehcnological should be technological; P11 L296 SE detector; P12 L 327 Fig 14. Should be Fig 15. Also, Fig 15a would benefit from arrows highlighting the cracks.; P12 L336. The “last layer” of the cladding is a little confusing, I suggest changing it to ‘initial layer’.; P13 L346 change is to was.; P13 L347 change to: The largest layer of hardening was…
Response 1: Thank you for your careful proofreading. All comments regarding spelling and grammar have been revised in the manuscript.
2. P13 L356-360 Final conclusion is fine, so can delete the last sentence.
Response 2:Thank you for the suggestion on the last conclusion. It is a general conclusion summarizing all the research results. This conclusion is not necessary, but we think it can be useful for potential readers.
3. "One point for the Authors to consider is the effect of the HAZ of the base steel may also be contributing to the wash out effect at the interface. I.e. Coarse Grain HAZ may have reduced toughness. As the research is based on interpretation of the cladding material, I don’t see it necessary to investigate this and it is also hard to eliminate the HAZ".
Response 3: Thank you for your valuable opinion. We fully agree with the suggestion that the condition of the base material may have a significant impact on the wash-out effect. We found microstructural changes in the HAZ zone of the base material. Due to the fact that the manuscript relates to the claddings produced by welding we have not developed this issue. However, we highlighted that this effect occurs at the border between steel and cladding. We explained this by the different mechanical properties of both materials (thus also the HAZ zones of the base material). Line: 281-282 "... This can be explained by the significant difference in hardness and mechanical behaviors of both materials (claddings and steel)."
If the answers are not exhaustive, we will gladly answer the next questions.
Thank you for your consideration!
Sincerely
Aleksandra Królicka
Reviewer 2 Report
The paper presents a very detailed characterization of the microstructure and wear mechanisms in claddings produced by welding, which are used to increase the wear strength in agricultural tools.
The text is well written and logically organized. The experimental analysis is comprehensive and makes the paper suitable for publication. I only found minor issues:
- Sec. 2: the base material and the cladding alloy seem not to be specified clearly;
- was the hardness determined by one or multiple measurements at one single location?
- Figure 4: not clear if the width of the bottom figures refers exactly to the location in the above hardness plot. If yes, please try to align better the above/below images so to make the images correspond more precisely to the hardness distribution;
- L85: "claddingsds" is "claddings"
- L127: "Hardens distribution" is "Hardness distribution" (?)
Author Response
Dear Reviewer,
Thank you for your review of our manuscript and a few crucial comments. We hope that with their help, we were able to improve the quality of the article. We have compiled your comments and our responses below:
1. Sec. 2: the base material and the cladding alloy seem not to be specified clearly;
Response 1: Due to the research purpose, the chemical composition is crucial, therefore it was included in the Results section, not in Materials and Methods (Line 111). Other detailed information on working conditions, the geometry of working elements, used materials were described in the previous work:
Kostencki, P.; Stawicki, T.; Królicka, A.; Sędłak, P. Wear of cultivator coulters reinforced with cemented-carbide plates and hardfacing, Wear, 2019, 438–439, 203063.
However, this manuscript provides information related to the assumed research purpose (important in this context). In our view, this has improved the readability of the text.
2. Was the hardness determined by one or multiple measurements at one single location?
Response 2: It is known that the claddings are characterized by some heterogeneity (e.g. layer thickness). A representative location was chosen for hardness measurements, which was also consistently used to microstructure characterization. The hardness measurement was made at appropriate distances between the indentations to obtain an accurate distribution. Of course, other areas were also checked, but the level of hardness was generally comparable.
3. Figure 4: not clear if the width of the bottom figures refers exactly to the location in the above hardness plot. If yes, please try to align better the above/below images so to make the images correspond more precisely to the hardness distribution;
Response 3: Thank you for your suggestion. However, we are afraid that adjusting micrographs to the thickness of the layers could cause less readability of the microstructures. Images were performed at the same magnification to show the difference in microstructure refinement. Thus, for example in the case of layer II (strongly refined), it would be impossible to identify the microstructure after adjusting the dimensions of the image. Therefore, the arrows indicate the appropriate layers.
4. L85: "claddingsds" is "claddings"
Response 4: Revised in the manuscript.
5. L127: "Hardens distribution" is "Hardness distribution"
Response 5: Revised in the manuscript.
If the answers are not exhaustive, we will gladly answer the next questions.
Thank you for your consideration!
Sincerely
Aleksandra Królicka